# Epigenetics of Skeletal Muscle-Associated Genes in the *ASB*, *LRRC*, *TMEM*, and *OSBPL* Gene Families

**Kenneth C. Ehrlich [1], Michelle Lacey [2,3] and Melanie Ehrlich [1,3,\*]**

[1] Center for Bioinformatics and Genomics, Tulane University Health Sciences Center, New Orleans, LA 70112, USA; kehrlich@tulane.edu

[2] Department of Mathematics, Tulane University, New Orleans, LA 70118, USA; mlacey1@tulane.edu

[3] Tulane Cancer Center, Tulane University Health Sciences Center, New Orleans, LA 70112, USA

[\*] Correspondence: ehrlich@tulane.edu; Tel.: +1-504-988-2449

**Abstract:** Much remains to be discovered about the intersection of tissue-specific transcription control and the epigenetics of skeletal muscle (SkM), a very complex and dynamic organ. From four gene families, Leucine-Rich Repeat Containing *(LRRC)*, Oxysterol Binding Protein Like *(OSBPL)*, Ankyrin Repeat and Socs Box *(ASB)*, and Transmembrane Protein *(TMEM)*, we chose 21 genes that are preferentially expressed in human SkM relative to 52 other tissue types and analyzed relationships between their tissue-specific epigenetics and expression. We also compared their genetics, proteomics, and descriptions in the literature. For this study, we identified genes with little or no previous descriptions of SkM functionality (*ASB4*, *ASB8*, *ASB10*, *ASB12*, *ASB16*, *LRRC14B*, *LRRC20*, *LRRC30*, *TMEM52*, *TMEM233*, *OSBPL6/ORP6*, and *OSBPL11/ORP11*) and included genes whose SkM functions had been previously addressed (*ASB2*, *ASB5*, *ASB11*, *ASB15*, *LRRC2*, *LRRC38*, *LRRC39*, *TMEM38A/TRIC-A*, and *TMEM38B/TRIC-B*). Some of these genes have associations with SkM or heart disease, cancer, bone disease, or other diseases. Among the transcription-related SkM epigenetic features that we identified were: super-enhancers, promoter DNA hypomethylation, lengthening of constitutive low-methylated promoter regions, and SkM-related enhancers for one gene embedded in a neighboring gene (e.g., *ASB8-PFKM*, *LRRC39-DBT*, and *LRRC14B-PLEKHG4B* gene-pairs). In addition, highly or lowly co-expressed long non-coding RNA (lncRNA) genes probably regulate several of these genes. Our findings give insights into tissue-specific epigenetic patterns and functionality of related genes in a gene family and can elucidate normal and disease-related regulation of gene expression in SkM.

**Keywords:** skeletal muscle; heart; myoblasts; enhancer; super-enhancers; DNA methylation; Leucine-rich Repeat; Oxysterol-Binding Protein-Like; Ankyrin Repeat and Suppressor of Cytokine Signaling Box; transmembrane protein

## 1. Introduction

Skeletal muscle (SkM), which is dispersed throughout the body, constitutes as much as 40% of the human body mass. It is a very complex organ composed of long cylindrical cells (myofibrils) containing up to hundreds of peripherally located nuclei and a specialized $Ca^{2+}$-storing form of the smooth endoplasmic reticulum (the sarcoplasmic reticulum; [1]). Myofibrils are organized in parallel rod-like myofibers that interface with nerve cells to permit voluntary muscle contraction. SkM is especially susceptible to physiological changes, e.g., exercise, which is accompanied by altered regulation of gene expression, changes in mitochondrial mass, and gross structural alterations in affected areas of the body [2,3]. In addition, given the multinucleated nature of SkM cells, they require activation of precursor (satellite) cells to form myoblasts (Mb) for the repair of damage to the

musculature. Loss of SkM mass without effective SkM repair is a major contributor to aging and disease, such as, sarcopenia, cachexia, muscular dystrophies, myotonia, and drug-induced myopathy.

The structural complexity and special functionality of SkM is matched by a large number of proteins specifically associated with SkM [1]. In turn, there are very many genes expressed specifically, or preferentially, in SkM. A transcriptomics (RNA-seq) comparison of human SkM to other tissue types indicated that hundreds of genes were enriched in expression in SkM [4]. The tissue type that is most similar to SkM is heart, which also contains striated muscle. However, more genes were enriched in SkM-only than were enriched in both SkM and heart, consistent with their differences in contractile function, developmental origin, and specific transcription factors (TFs). There is some heterogeneity in transcriptional profiles between different types of SkM [5] but differences in gene expression between SkM and non-SkM tissues are much greater [1,5,6].

SkM differentiation, homeostasis, and reprogramming, such as fiber-type switching, relies on SkM-specific regulation of gene expression through SkM lineage-associated TFs, cell signaling, and epigenetics [7–12]. The TF drivers of myogenic differentiation are the four myogenic regulatory TFs encoded by *MYOD1*, *MYF5*, *MYF6*, and *MYOG*, whose expression is cell type-specific and regulated by promoters, enhancers, and DNA hypomethylation [13–16]. There is chromatin remodeling throughout the genome, including formation and maintenance of SkM lineage-specific enhancers associated with myogenesis [17] or homeostasis [18] set up, in part, by binding of MYOD [19]. In addition, SkM lineage-specific hypomethylation, less extensive hypermethylation [8,10], and 5-hydroxymethylation [11] of DNA are seen. Moreover, SkM-specific repressor chromatin [20], non-coding RNAs [9], changes in signal transduction pathways [11,21], and post-translational modifications of non-histone proteins, including MYOD1 [1,22], are essential to forming and maintaining SkM.

Epigenetics plays critical roles in tissue-specific gene expression [23–25]. Despite the importance of SkM to health, its distinctive developmental program, and its special tissue organization, there is much that is not yet understood about the SkM-associated epigenetics of genes with known muscle-specific functions. Moreover, there are still some poorly characterized genes that are preferentially expressed in human SkM but whose specific functions in the SkM lineage have not been delineated [4]. In our study, we used a comprehensive human RNA-seq database to identify little-studied genes that have much higher expression in SkM than in most or all other tissues. For detailed bioinformatics analyses (transcriptomics, epigenetics, protein similarity analysis, and literature searches), we then chose eleven of these genes from the following four gene families or super-families: Leucine-Rich Repeat Containing *(LRRC)*, Oxysterol Binding Protein Like *(OSBPL)*, Ankyrin Repeat and Socs Box *(ASB)*, and Transmembrane Protein *(TMEM)*. We compared these to other genes with known SkM functions in these four gene families to gain insight into the functionality of the little-studied genes and into tissue-specific epigenetics relative to tissue-specific transcription. Our study helps elucidate the variety of tissue-specific promoter, enhancer, and DNA hypomethylation profiles that closely correlate with relative transcription levels among different tissues and cell types for these four families of genes.

## 2. Results

### 2.1. The SkM-Related Genes and Their Gene Families Chosen for Epigenetic Analysis

We searched for protein-coding genes preferentially expressed in SkM among the 56,202 genes in the GTEx RNA-seq database [6], which has hundreds of biological replicates for most of the 53 tissue types. We required at least five times more transcripts per million (TPM) in SkM than in the median of the 52 other tissues and a SkM TPM of >5. We further stipulated that the ratio of TPM in SkM to that in heart (left ventricle) was ≥5 and that the same preference held for SkM versus aorta (Tables S1 and S2). The genes that met these criteria are referred to as SkM genes. For our study, we chose four families or super-families (*ASB*, *LRRC*, *OSBPL*, and *TMEM*) containing at least one SkM gene that had no previous description of their relationship to SkM or only brief mentions in the

literature (Tables S3 and S4a). These genes are *ASB4*, *ASB8*, *ASB12*, *ASB16*, *LRRC20*, *LRRC30*, *OSBPL6*, *OSBPL11*, *TMEM52*, and *TMEM233*. We compared the tissue-specific epigenetics, transcriptomics, and sequence conservation of these eleven SkM genes to each other, to those of four other better studied SkM genes in these families (*ASB2*, *ASB5*, *LRRC38*, and *TMEM38A*), and to seven genes that are preferentially expressed in both SkM and heart (*ASB10*, *ASB11*, *ASB15*, *LRRC2*, *LRRC14B*, *LRRC39*, and *TMEM38B*). Genes preferentially expressed in SkM and heart are referred to as SkM/heart genes and met similar criteria for overall SkM selective expression but had a ratio of TPM in SkM to TPM in heart of 1.5–3. The 21 examined genes had much lower expression in aorta (a tissue rich in smooth, non-striated muscle) than in SkM or heart (Table S4a). Six of the studied genes have been associated with cancer, two with cardiac disease (*LRRC2* and *LRRC14B*), one with Emery–Dreifuss muscular dystrophy (*TMEM38A*), one with myotonic dystrophy type 1 (*ASB2*), one with glaucoma (*ASB10*), one with Treacher Collins syndrome (a craniofacial disease; *OSBPL11*), and one with osteogenesis imperfecta (*TMEM38B*; Table S6).

### 2.2. LRRC Genes that are Preferentially Expressed in SkM Display Muscle-Associated Enhancers or Super-Enhancers Containing Hypomethylated DNA Regions

Three SkM and three SkM/heart genes are among the 67 genes in the *LRRC* (Leucine-Rich Repeat Containing) superfamily, and these six genes displayed RNA steady-state levels in SkM (from the median of 803 SkM samples) that were ~14 to >500 times higher than the median TPM for the other GTEx tissues (Table S4a). The median TPM from all the non-SkM samples was obtained using median values for each tissue obtained, usually from >200 biological replicates each (Table S4). Among these six genes, we found little or no literature about *LRRC14B*, *LRRC20*, and *LRRC30* in SkM or muscle, unlike the case for *LRRC2*, *LRRC38*, and *LRRC39*. Consistent with their specific expression in SkM or in both SkM and heart, these genes display regions of strong SkM or SkM/heart intragenic enhancer chromatin, as seen, for example, for *LRRC38* and *LRRC14B* (Figure 1, orange in chromatin state tracks). Chromatin states were determined by the Roadmap Epigenomics Consortium [23] from genome-wide profiles of histone H3 lysine-4 trimethylation (H3K4me3) and H3 K27 acetylation (H3K27ac) for promoter chromatin and H3K4me1 and H3K27ac for enhancer chromatin (18-State/Auxiliary Hidden Markov Model). The enhancer chromatin in *LRRC38*, *LRRC14B*, and *LRRC20* in SkM qualifies as a super-enhancer, a cluster of strong enhancer (and promoter, red colored) chromatin regions, which can be detected by tissue-specific enrichment in histone H3 lysine-27 acetylation (H3K27ac) over a >5 kb region [26,27]. With the exception of heart and adrenal gland, as described below, the non-SkM tissues examined for correlating tissue-specific epigenetics with transcription in Figures 1 and 2 had no overlap of their interquartile range of TPM values with that of SkM (Figures S1 and S2, Table S4a).

SkM DNA hypomethylation was associated with sub-regions of enhancer chromatin for all of the SkM or SkM/heart *LRRC* genes (Table 1). We determined significant SkM-specific differentially methylated regions (DMRs) from bisulfite-seq (BS-seq) data [23] for SkM versus for heart, aorta, lung, adipose tissue, and monocytes using stringent criteria (see the Methods Section). Although *LRRC39* did not have a significant hypomethylated DMR in the gene body or intergenic upstream region, it did exhibit a SkM-associated low-methylated region (LMR, blue horizontal lines in bisulfite-seq tracks) in its promoter region (Figure 2d, green arrow). An LMR is a region of significantly low methylation relative to the rest of the same genome [28]. There was also a SkM and heart hypomethylated DMR (Figure 2d, orange arrow and data not shown) overlapping a SkM and heart LMR at the 3′ end of the *LRRC39*-adjacent gene, *DBT* (dihydrolipoamide branched chain transacylase E2). This DMR was located in a ~1.6-kb region of strong enhancer chromatin observed in SkM and heart 10 kb upstream of the *LRRC39* transcription start site (TSS). While *LRRC39* codes for a highly tissue-specific component of SkM and heart sarcomeres [29], *DBT* encodes a broadly expressed subunit of an inner mitochondrial enzyme. Given the lack of SkM and heart specificity of *DBT* expression (Figure 2d; Table S4b), it is likely that this *DBT*-overlapping SkM/heart hypomethylated enhancer acts on the *LRRC39* promoter and not on the *DBT* promoter in striated muscle. The liver LMR and small enhancer chromatin region found at the same *DBT* intragenic position might help drive the slightly higher expression of *DBT* in liver relative to that of most tissues (liver TPM, 6.5,

versus median tissue TPM, 5.8) but is unlikely to impact *LRRC39* expression in liver, which is low (liver TPM, 1.6, versus median tissue TPM, 3.3). This provides an interesting example of a tissue-specific DNA-hypomethylated enhancer region that is likely to be regulating different genes in different tissues. *TRMT13*, which overlaps the 3′ end of *LRRC39*, has RNA levels for SkM and heart that are among the lowest for this broadly expressed gene (Table S4b). Unlike the 5′ upstream gene to *LRRC39*, *TRMT13* lacks SkM-associated enhancer chromatin (data not shown).

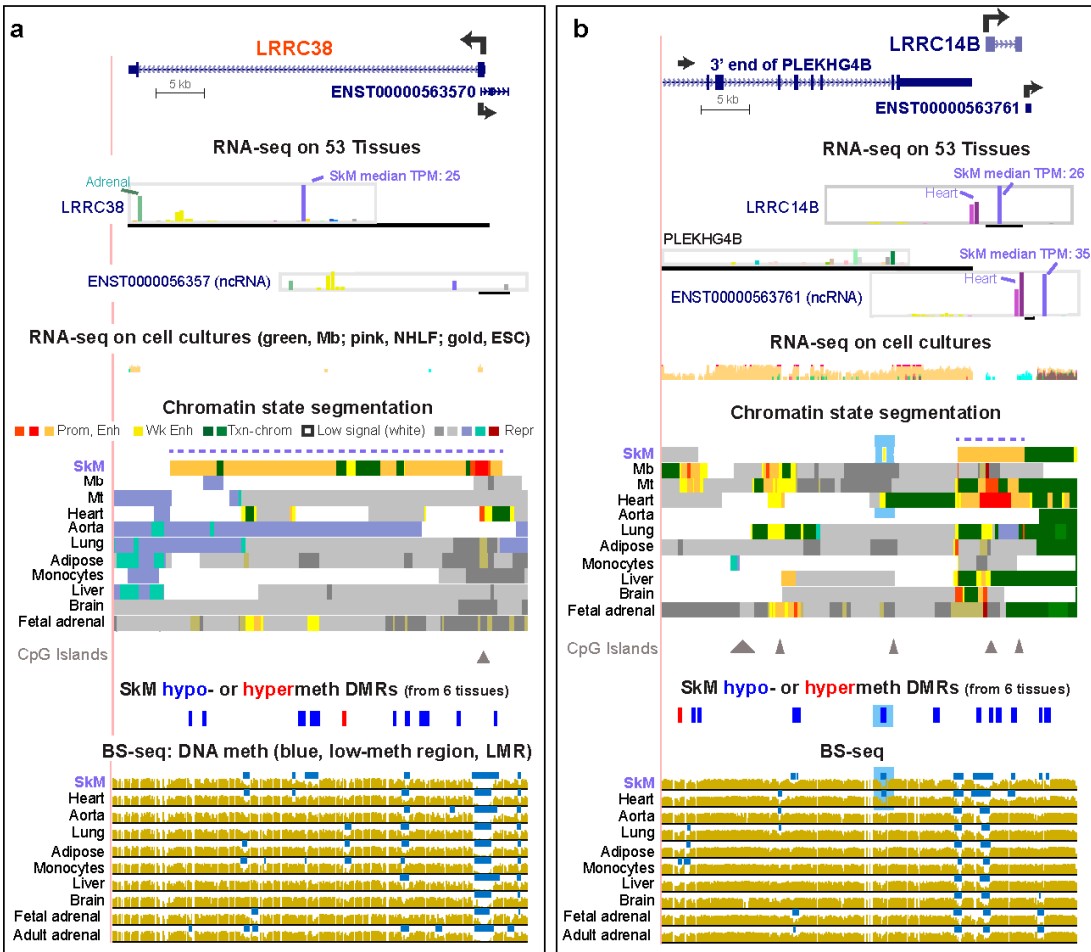

**Figure 1.** Enhancer and DNA methylation profiles are consistent with the tissue-specific expression of *LRRC38* and *LRRC14B*. (**a**) *LRRC38* (chr1:13,799,705-13,844,788) RefSeq isoform and an AS ncRNA gene at its 5′ end. (**b**) *LRRC14B* (chr5:157,091-201,293) and a neighboring ncRNA gene and protein-encoding gene. For both panels, GTEx RNA-seq expression profiles are shown as linear transcripts per million (TPM) bar graphs. RNA-seq for cell cultures (Mb, myoblasts, green; NHLF, normal human lung fibroblasts, pink; ESC, embryonic stem cells, gold) is given as a composite overlaid signal. Chromatin state segmentation denotes predicted promoter chromatin (prom; light or bright red as weak or strong promoter, respectively), strong enhancer (enh; orange), weak enhancer (wk enh; yellow), or repressed chromatin (rep: dark or light gray, enriched in H3K27me3; light blue or blue-green, enriched in H3K9me3; brown, bivalent promoter enriched in H3K27me3 and H3K4me1), or H3K36me3-enriched transcribed chromatin (txn chrom, green). Triangles, CpG islands. Significant hypomethylated (blue) or hypermethylated (red) DMRs in skeletal muscle (SkM) versus heart, aorta, lung, adipose, or monocytes were determined from bisulfite-seq (BS-seq). Blue bars in BS-seq, low-methylation relative to the same genome. All tracks are from hg19 in the UCSC Genome Browser and are aligned. The DMR track is the only custom track. Dotted lines above chromatin state tracks, SkM super-enhancers; blue highlighting in panel **b**, weak enhancer chromatin specific to SkM and heart that overlaps SkM and heart DMRs; red font for gene names, genes that are preferentially expressed in SkM but not in heart.

**Table 1.** Epigenetic patterns associated with transcription in SkM among the Ankyrin Repeat and Socs Box (ASB), Leucine-Rich Repeat Containing (LRRC), Oxysterol Binding Protein Like (OSBPL), and Transmembrane Protein (TMEM) family genes [a].

| Epigenetic Patterns Associated with SkM | *ASB* Family (9 Genes) | *LRRC* Family (6 Genes) | *OSBPL* Family (2 Genes) | *TMEM* Family (4 Genes) |
|---|---|---|---|---|
| Prom chrom assoc with expression in SkM | *ASB2, 4, 5, 10, 11, 15* | *LRRC2, 14B* [b]*, 20, 30, 38,39* | none | *TMEM52, 233* |
| SkM-assoc prom DNA hypometh | *ASB4, 5, 10, 11, 12, 15* | *LRRC30, 39* | none | none |
| constit LMR at TSS | *ASB2, 8, 16* | *LRRC2, 14B, 20, 38* | *OSBPL11* | none |
| CpG island at prom | *ASB8* | *LRRC14B, 20, 38* | *OSBPL6, 11* | *TMEM38A, 38B, 52* |
| Super-enhancer | *ASB2,5, 8, 11* | *LRRC14B, 20, 38* | none | *TMEM52* |
| SkM enh chrom in adjacent gene | *ASB2, 8, 10, 16* | *LRRC14B, 39* | none | *TMEM52; 38A* |
| Intragenic enhancer chrom | *ASB2, 4, 5, 8, 10, 11, 12, 15, 16* | *LRRC2, 14B, 20, 38, 39* | *OSBPL6, 11* | *TMEM38A, 38B, 52, 233* |
| Intergenic proximal enh chrom | *ASB2, 5, 8, 10, 12, 15, 16* | *LRRC14B, 30, 38* | *OSBPL11* | *TMEM52* |
| Intergenic distal enh chrom | *ASB2, 4, 10, 15, 16* | *LRRC14B, 30* | *OSBPL11* | *TMEM38A, 38B, 233* |
| DNA hypometh in SkM-assoc enh chrom | *ASB2, 5, 10, 11, 12, 15, 16* | *LRRC2, 14B, 20, 30, 38, 39* | *OSBPL6, 11* | *TMEM38A, 38B, 52, 233* |

[a] The epigenetic features for these 21 genes are shown in Figures 1–5 and Figures S1–S3, except for *ASB11* and *ASB15*. Abbreviations: chrom, chromatin; assoc, associated; prom, promoter (H3K4me3/H3K27ac-enriched); enh, enhancer (H3K4me1/H3K27ac-enriched), constit, constitutive (present in all or almost all tissues); LMR, low methylated region (determined by bisulfite-seq profiles relative to that of the given tissue's genome [28]); intragenic enh chromatin, enhancer chromatin overlapping the gene body; intergenic proximal enh chrom, enhancer chromatin upstream of and adjacent to the promoter region; intergenic distal enh chromatin, enhancer chromatin far upstream of the promoter region. [b] The active chromatin over the *LRRC14B* promoter specifically in SkM was designated as enhancer chromatin in the Roadmap database but it was enriched in H3K4me3, indicative of active promoter activity, as well as H3K4me1, reflecting active enhancer activity.

The other SkM or SkM/heart *LRRC* genes with known functionality in muscle are *LRRC38*, a BK channel protein [30], and *LRRC2*, a gene previously associated with heart hypertrophy and mitochondria [31] and downregulated in SkM upon stabilized weight-loss in human SkM [32]. *LRRC2* RNA has a ~2.4 fold higher steady-state level in SkM than in heart (Table S4a). That this gene has its highest expression in SkM is consistent with the downstream extension of its constitutive LMR (Figure 2b, bisulfite-seq tracks, blue arrow) at the CpG island-promoter in this tissue. A small promoter chromatin region in the middle of intron 1 in *LRRC2* overlaps the antisense (AS) gene *LRRC2*-AS1, which has RNA levels that are very low in many tissues, highest in testis (TPM, 0.63), and extremely low in SkM (TPM, 0.1). The *LRRC2-AS1* transcript (*ENST00000599511*) is predicted to encode a protein with 171 amino acids (aas) but without any conserved domains that would allow for its functional assessment. It might act in trans as a micro-peptide encoded by a "lncRNA" [33] in some cell types. In H1 embryonic stem cells (ESC), *LRRC2-AS1* was preferentially expressed (Figure 2c, oval). Promoter chromatin and a DNaseI-hypersensitive site overlap this AS gene in most cell and tissue types (Figure 2b and data not shown). These epigenetic marks might reflect a chromatin structural function for this very weakly expressed AS gene in most cell types and tissues because they are superimposed on a constitutive binding site for the chromatin looping protein CCTC-binding factor (CTCF; data not shown [34]). Another AS gene specifying a non-coding transcript (*ENST0000056357)* overlaps the 5' end of *LRRC38* (Figure 1a). Both the sense and AS *LRRC38* genes display preferential expression in SkM and the adrenal gland, suggesting that this promoter-overlapping head-to-head lncRNA gene is involved in positive cis-regulation [33] of *LRRC38*. The preferential expression of *LRRC38* in the adult adrenal gland (for which chromatin state data are not available) as well as in SkM is reflected in the distinctive tissue-specific LMRs for both of these tissues (Figure 1a, bottom).

The SkM/heart-specific super-enhancer in the little-studied SkM/heart gene, *LRRC14B* (Figure 1b, dotted line in chromatin tracks), overlaps SkM and heart hypomethylated DMRs. Further upstream of *LRRC14B* in the 3′ end of *PLEKHG4B* (Pleckstrin Homology and RhoGEF Domain containing G4B) is a hypomethylated SkM/heart DMR residing in SkM/heart weak-enhancer chromatin (Figure 1b, blue highlighting). *PLEKHG4B* has negligible levels of expression in SkM or heart. Therefore, this enhancer region is likely to be associated with *LRRC14B* expression in SkM rather than with transcription of the broadly expressed *PLEKHG4B* (Table S4a), making it one of the eight genes examined in this study with one of its enhancers apparently lodged in a neighboring gene (Table 1). In addition, immediately downstream of and in the sense direction to *LRRC14B*, a lncRNA gene is seen that has remarkably high levels of transcription in SkM and heart (Figure 1b). The levels of that transcript (*ENST0000056376*) are even higher than those of *LRRC14B* and share a similar expression profile with it (Figure 1b and Table S4b). This 564 bp gene overlaps enhancer chromatin in the heart and is adjacent to enhancer chromatin in SkM. About 22 kb downstream of *LRRC14B* is *SDHA* (not shown in Figure 1b; Succinate Dehydrogenase Complex Flavoprotein subunit A), which, although expressed highly in most tissues, has its highest expression in SkM and heart (TPM, 238 and 302, respectively; Table S4b). Gene enhancer regulatory elements [34,35] predict a conformational association between *SDHA*, *ENST00000563761*, and *LRRC14B* (data not shown).

*LRRC30*, a 0.9 kb SkM gene with no introns and of unclear functional significance, is only modestly expressed in SkM (TPM, 6). However, it is not expressed in any of the other 52 different types of tissues or >10 types of cell cultures that were examined, with the exception of extremely low expression in testis (TPM, 0.1; Table S3; [6]). It displays highly SkM-specific promoter chromatin extending beyond the gene, which was bordered by enhancer chromatin and overlapped DNA hypomethylation in SkM (Figure 2).

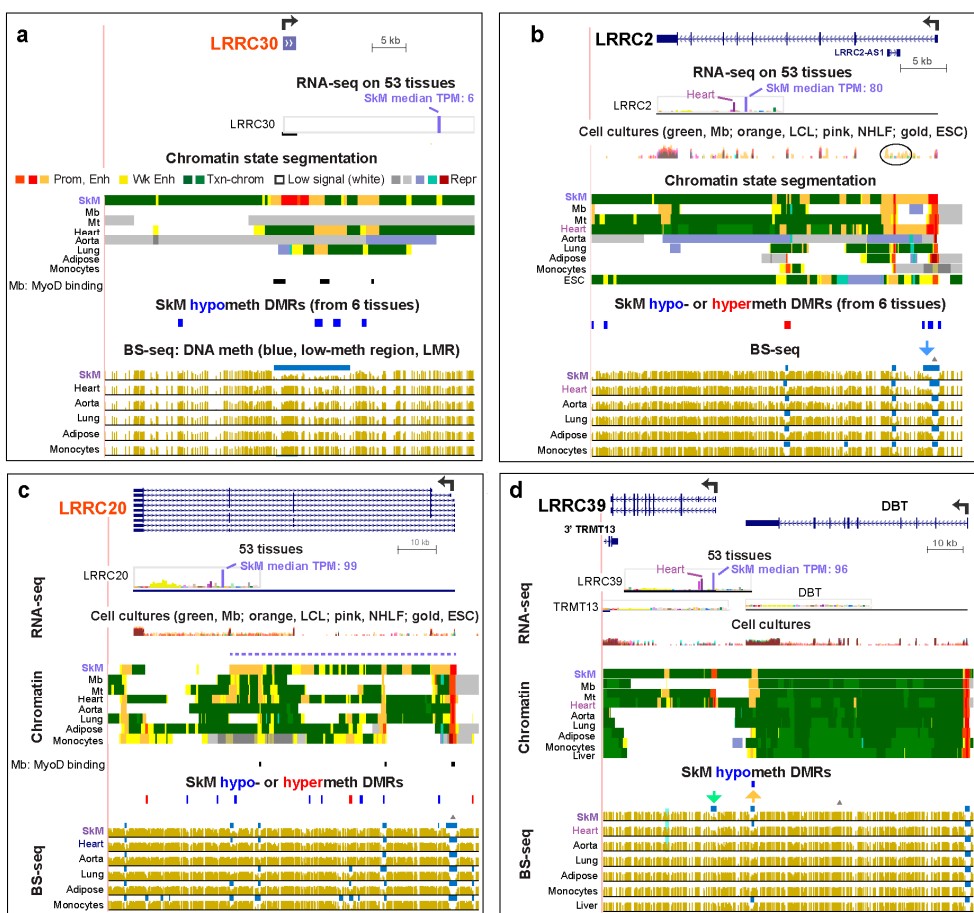

**Figure 2.** SkM or SkM/heart-specific DNA hypomethylation and enhancer or promoter chromatin at *LRRC30, LRRC2, LRRC20,* and *LRRC39* were found within the gene body, upstream, downstream, or

in the adjacent gene. (**a**) *LRRC30* (chr18:7,218,000-7,245,179), (**b**) *LRRC2* (chr3:46,545,011-46,612,516), (**c**) *LRRC20* (chr10:72,052,172-72,148,755), and (**d**) *LRRC39* (chr1:100,611,823-100,717,038). Notations are as in Figure 1 except for the added MYOD binding sites orthologous to those identified in mouse myoblasts (Mb) and myotubes (Mt) (C2C12) by chromatin immunoprecipitation (Myod-ChIP; [36]) that were found in or around *LRRC30* and *LRRC20*. Only binding sites with scores ≥40 are shown. Triangles above the BS-seq tracks are CpG islands. Blue and orange arrows and blue highlighting in panels **b** and **d** are described in the text. RNA-seq for cell cultures is shown in panels **b–d**.

In contrast, *LRRC20*, an 84 kb SkM gene that has also received little attention, is expressed at considerable levels in many tissues, especially in brain, testis, and some smooth muscle-rich tissues, but at much higher levels in SkM (Table S4a) with concomitantly more intragenic enhancer chromatin (Figure 2c).

We also compared available expression and chromatin state data for these *LRRC* genes in cultured myoblasts (Mb), their multinucleated differentiation product myotubes (Mt), and diverse normal non-myogenic cell cultures [23]. *LRRC2*, *LRRC14B*, and *LRRC39* displayed much higher expression in Mb or Mt than in other cell cultures (Table S5a) with the exception of *LRRC2* in ESC (Table S5). These genes in Mb or Mt either shared enhancer chromatin with SkM or had unique enhancer chromatin regions (Figures 1b, 2b, and 2d).

### 2.3. Preferential Expression of OSBPL6 and OSBPL11 in SkM was Associated with SkM-Specific Enhancer Chromatin Mostly within the Gene Body or Upstream

*OSBPL6* (*ORP6*) and *OSBPL11* (*ORP11*) in the 10-member *OSTBPL/ORP* (Oxysterol-Binding Protein-Like) gene family, are expressed at the highest levels in SkM, although they are widely expressed at lower levels in most other tissues (Figure 3, Table S4a). *OSBPL* family members are intracellular lipid receptors that generally bind oxysterols or cholesterol [37]. The specific SkM functionality of OSBPL6 and OSBPL11 proteins is not known nor is the identity of their lipid ligand, although they have been reported to be localized to the nucleus as well as the cytosol, plasma membrane, and endoplasmic reticulum (OSBPL6), or cytosol and golgi apparatus (OSBPL11) [34,37]. The greater number or extent of enhancer chromatin regions in SkM than in other tissues is in accord with their tissue-specific expression profiling (Figure 3 and Figure S3). *OSBPL6* in SkM exhibits many intragenic discontinuous regions of SkM-specific enhancer chromatin. The LMRs in SkM were in different intragenic locations from those in skin (Figure 3c, blue bars in bisulfite-seq tracks; chromatin state data are not available for skin), which was among the tissues with the next highest levels of *OSBPL6* expression. One of the distal promoter-associated splicing isoforms, variant #5 (*ENST00000359685*), predominates in SkM, and only ~16% as much RNA signal originates from a far-proximal promoter [6]. In contrast, frontal cortex and cerebellum had variant 4 (*ENST00000190611*), which encodes as the main isoform. This variant originates from the same distal promoter as #5 but differs in its splicing pattern and gives a protein that is 36 aa longer [6].

*OSBPL11* had more expression in the non-SkM samples than did *OSBPL6* (Table S4a). This is reflected in the high levels of intragenic transcription-associated H3K36 trimethylation (H3K36me3) in the gene body of *OSBPL11* in all of the examined tissues and cell cultures, including Mb and Mt (Figure 3b, green color in chromatin state tracks), although SkM had the highest expression. Also, unlike *OSBPL6*, *OSBPL11* has most of its SkM-specific enhancer and hypomethylated DMRs extending up to 29 kb upstream of its TSS (Figure 3b, d). *OSBPL11* in liver has a hypomethylated upstream enhancer with an embedded strong promoter subregion (Figure 3d, pink arrow) that may help drive the moderate levels of *OSBPL11* RNA in liver (TPM, 7.9), even though liver steady-state RNA levels are lower than the median (TPM, 10.4).

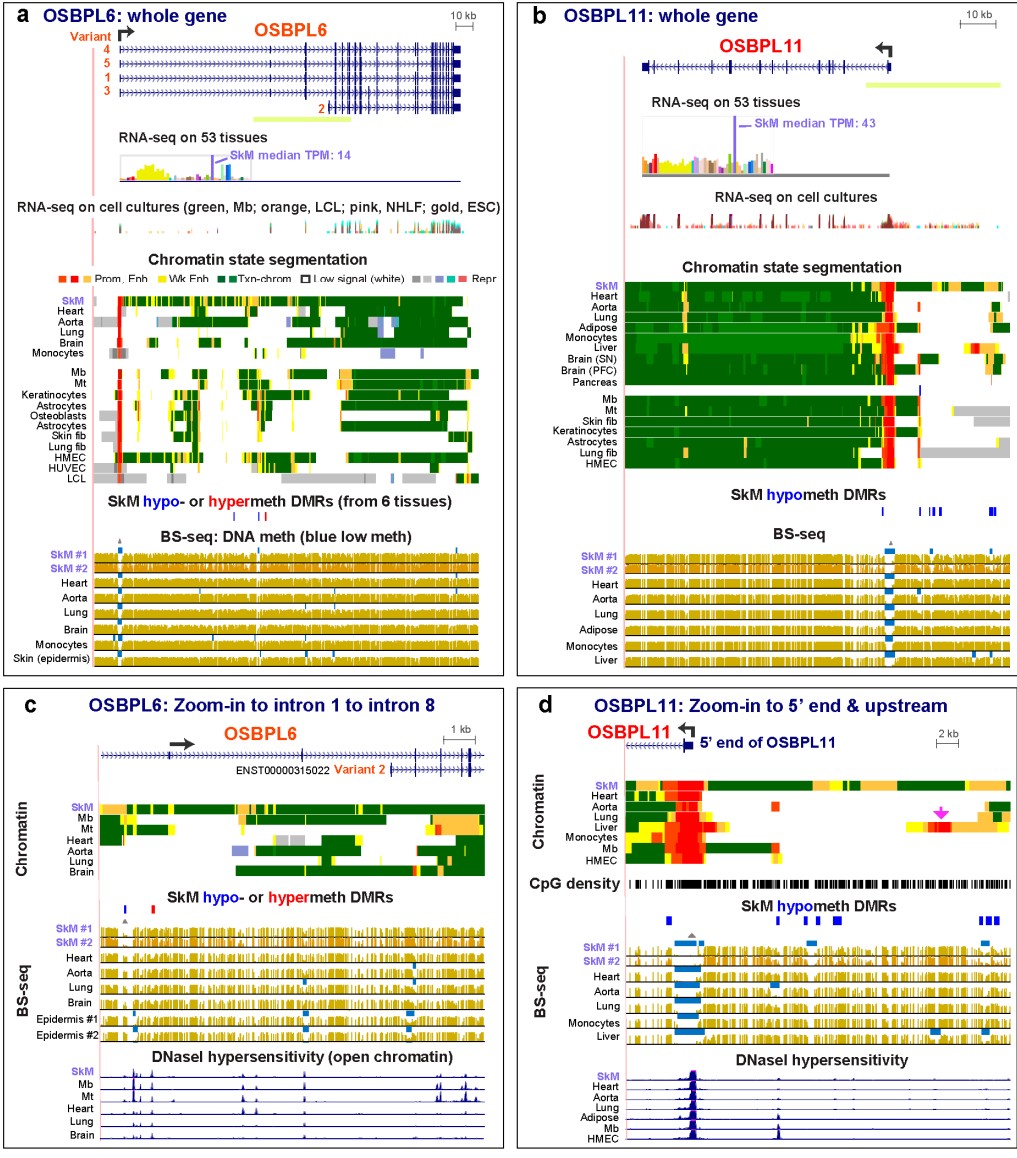

**Figure 3.** *OSBPL6* and *OSBPL11*, genes expressed preferentially in SkM, have scattered regions of SkM-specific enhancer chromatin that differ from those of myoblasts and myotubes. (**a**) *OSBPL6* with its five RefSeq isoforms (chr2:179,043,977-179,275,410). (**b**) *OSBPL11* (chr3:125,243,153-125,346,217). (**c**) A central portion of the *OSBPL6* gene (yellow-green line in panels a and b) shown at higher magnification (chr2:179,138,760-179,199,967). (**d**) The 5' region of *OSBPL11* (chr3:125,308,218-125,343,535). Brain PFC, brain prefrontal cortex; brain SN, brain substantia nigra; Fib, fibroblast; HMEC, human mammary epithelial cell line; HUVEC, human umbilical vein endothelial cell strain; SkM #2, second SkM sample from unknown location. Unless otherwise noted, SkM, refers to SkM #1. UCSC Genome Browser tracks were as in the previous figures except that DNaseI hypersensitivity was added. Light green bars under gene figures in panels (**a**) and (**b**) are the regions shown in higher resolution in panels (**c**) and (**d**).

### 2.4. Nine of the 18 ASB Genes Display SkM or SkM/Heart Preferential Expression that was Associated with Loss of DNA Methylation in or Adjacent to the Promoter

Among the 18 genes of the *ASB* (Ankyrin Repeat and Suppressor of Cytokine Signaling Box) gene family, which encode ubiquitin E3 ligases [38], are six SkM genes (*ASB2*, *ASB4*, *ASB5*, *ASB8*, *ASB12*, and *ASB16*) and three SkM/heart genes (*ASB10*, *ASB11*, and *ASB15* (Table S4a)). Five (*ASB4*, *ASB8*, *ASB10*, *ASB12*, and *ASB16*) have not been described in previous publications as to the cellular roles of their ubiquitin ligase activity in striated muscle [39]. *ASB4* is the only one of these nine genes that had

higher expression in a few non-muscle tissues than in SkM, namely, in adrenal gland and pituitary, although RNA levels in SkM were higher than in the other 50 examined tissues (Figure 4b). Similarly, for the tissue types shown in Figure 4, the interquartile range of TPM values for SkM was higher than those of the other examined tissue types with no overlaps except for *ASB4* in SkM and pituitary (Figure 4b and Figure S4). Epigenetic data, which were available for the adrenal gland but not pituitary, indicated more DNA hypomethylation and enhancer chromatin at *ASB4* in the adrenal gland than in SkM, consistent with the observed expression differences. Preferential expression of the above nine *ASB* genes in SkM was accompanied by intragenic enhancer chromatin and DNA hypomethylation (Figure 4, Figures S5–S7, and data not shown). Four of these genes were embedded in a tissue-specific super-enhancer in SkM (Table 1; Figure 4c,d). Three *ASB* genes displayed constitutive LMRs at the promoter region that were extended downstream or upstream in SkM (Figure 4a, d and Figure S6; blue arrows over bisulfite-seq tracks). Six other *ASB* genes had LMRs specifically in SkM and other expressing tissues (Figure 4b, c and Figure S5; Table 1). Profiles of DNaseI hypersensitivity and CTCF binding were generally less informative of SkM or Mb preferential expression for the studied genes than were the chromatin state and DNA methylation profiles (e.g., Figures S5–S7).

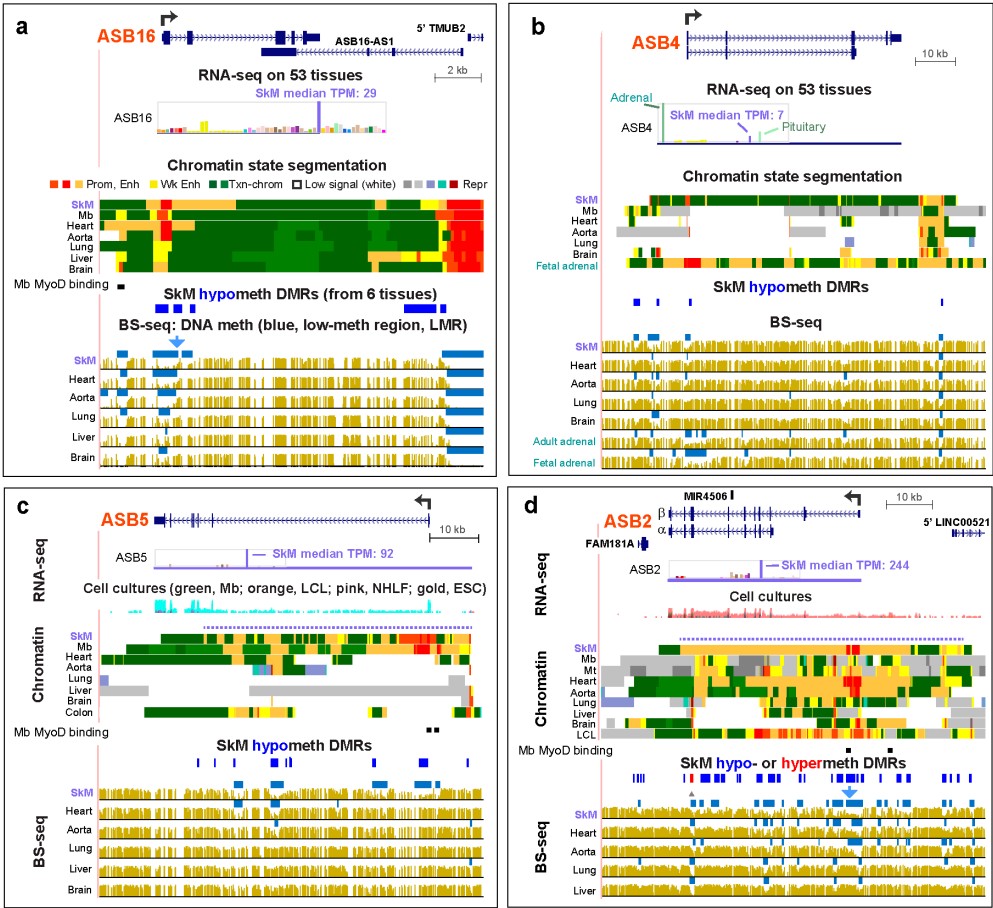

**Figure 4.** The relative expression levels in SkM of *ASB16*, *ASB4*, *ASB5*, and *ASB2* correspond with their extent of SkM-specific DNA hypomethylation at their promoters. (**a**) *ASB16* (chr17:42,244,742-42,265,156). (**b**) *ASB4* (chr7:95,093,590-95,190,891). (**c**) *ASB5* (chr4:177,123,672-177,201,191). The chromatin state profile for Mt, which is not shown, was similar to that of Mb. (**d**) *ASB2* (chr14:94,385,586-94,470,905). MyoD binding was seen for *ASB16*, *ASB5*, and *ASB2* and is shown as described for Figure 2. Colon, smooth muscle layer of colon.

SkM genes *ASB5* (implicated in regulating muscle mass [38]) and *ASB12* (a little-studied gene) are highly and specifically expressed in SkM, except that *ASB5* also exhibited moderate steady-state RNA levels in several smooth muscle-rich tissues, including colon (Table S4a). The levels of *ASB5* RNA in the

colon were matched by TSS-far downstream enhancer chromatin in that tissue, although transcription in the colon might begin at an upstream promoter (Figure 4c). Heart, which had very low levels of *ASB5* RNA and *ASB12* RNA (Figures 5, S4 and S5), displayed some intragenic enhancer chromatin and active-transcription-type chromatin (enrichment in H3K36me3). However, in the heart, these genes either lacked promoter-type chromatin (*ASB5*, Figure 4c) or had only weak promoter chromatin, as seen in the H3K27ac and H3K4me3 signal profiles (*ASB12*; Figure S5 and data not shown). The lack of Mb and Mt RNA for *ASB12* (Table S5a) and the absence of promoter or enhancer chromatin at this gene in these cells (Figure S5) suggest that this gene has a very specific role to play in SkM tissue, per se, and one that is associated with only moderate amounts of gene expression (SkM TPM, 16). In contrast, *ASB5* had the highest expression in Mb and Mt of any studied gene, negligible expression in non-myogenic cell cultures, and high levels of expression in SkM (SkM TPM, 92).

*ASB16* had a large enhancer chromatin region extending from upstream of the TSS to the beginning of intron 2 with overlapping SkM hypomethylation (Figure 4a). Near the 3′ end of the gene is the 5′ end of a large AS gene (*ASB16-AS1*) that is broadly expressed and does not exhibit the SkM specificity of *ASB16*, and apparently has a function independent of that of *ASB16* [40]. Interestingly, the long enhancer chromatin region surrounding strong promoter chromatin at the *ASB16* TSS in the heart was accompanied by only low steady-state levels of heart *ASB16* RNA (Table S4a). This might indicate post-transcriptional downregulation of *ASB16* RNA levels in the heart.

*ASB8* and *ASB10* display SkM or SkM/heart hypomethylated DMRs in their nearest neighbor genes, *PFKM* (Phosphofructokinase, Muscle) and *IQCA1L* (IQ Motif with AAA Domain 1 Like), respectively (Figures S6 and S7). While *IQCA1L* is a testis-specific gene, *PFKM* is broadly expressed, except with much higher levels of RNA in SkM (TPM, 503) than in any other tissue (Table S4b), which is in accord with a muscle-cramping phenotype that is associated with certain inherited mutations in this gene [34]. *PFKM*'s distribution of enhancer chromatin in SkM indicates a shared *ASB8*/*PFKM* SkM super-enhancer (Figure S6b). Unlike SkM, Mb and Mt did not express *ASB10* but did express *ASB8*, although not specifically (Table S5a).

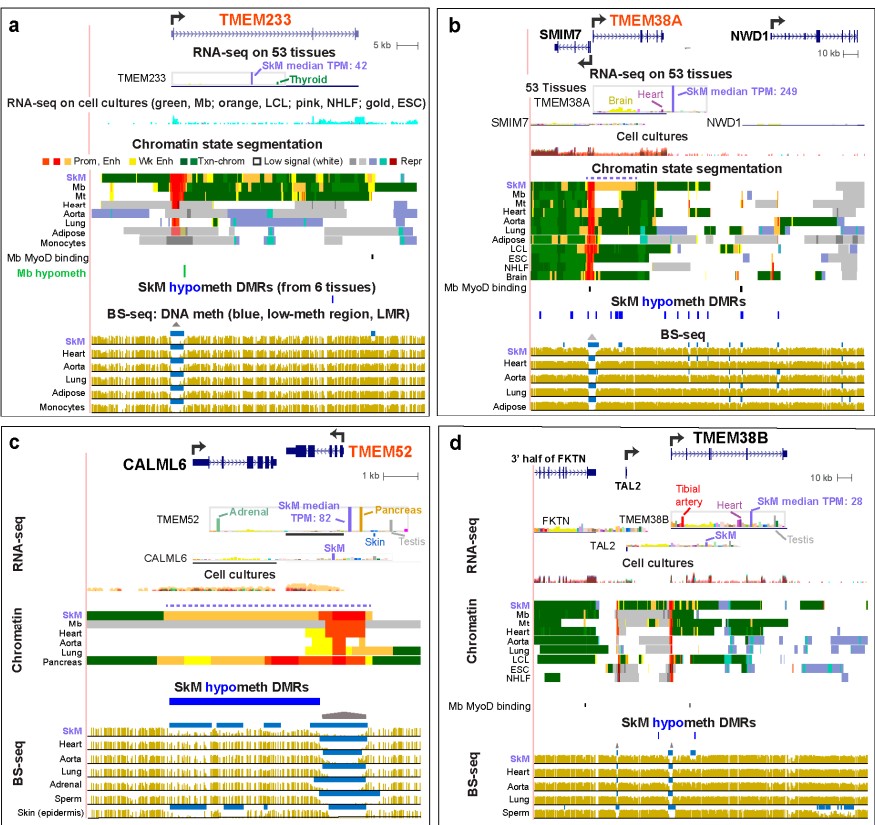

**Figure 5.** Preferential transcription of *TMEM233*, *TMEM38A*, *TMEM52*, and *TMEM38B* in SkM was accompanied by different distributions of SkM-specific enhancer chromatin. (**a**) *TMEM233*

(chr12:120,010,669-120,096,484). (**b**) *TMEM38A* (chr19:16,747,743-16,878,746). There are several isoforms for *SMIM7*, only one of which is displayed. (**c**) *TMEM52* and *CALML6*, which is also preferentially expressed in SkM (chr1:1,843,125-1,853,017). (**d**) T*MEM38B* and *TAL2*, which has its highest expression in SkM and pituitary (TPM, 0.7) among tissues and in Mt (FPKM, 11; chr9:108,359,776-108,596,109). The TPM scale for *TAL2* was decreased to make the bar graph results more visible (see Figure S4).

*2.5. SkM-Associated Genes in the TMEM Family Exhibited SkM-Specific Intragenic, Intergenic, or Neighboring-Gene Enhancer Chromatin*

Genes in the *TMEM* (Transmembrane Protein) super-family constitute the largest group of protein-encoding genes that we considered and have various types of transmembrane domains with diverse functions. Of the more than 250 genes designated as "*TMEM*", only four (*TMEM233*, *TMEM52*, *TMEM38A*, and *TMEM38B*) are preferentially transcribed in SkM or SkM and heart. The little-studied SkM genes, *TMEM233* and *TMEM52*, have no descriptive gene summary in Entrez Gene and very little literature about them (Tables S4a and S6). In contrast, *TMEM38A* (*TRIC*-A) and *TMEM38b* (*TRIC*-B) are known to encode proteins involved in ion channeling in the sarcoplasmic reticulum [41]. TMEM38A protein has also been shown to redirect specific sets of genes to the nuclear envelope during myogenesis [42]. Mutations in *TMEM38B* cause one subtype of osteogenesis imperfecta, which can include SkM and cardiac deficiencies [43].

*TMEM233* is highly and specifically expressed in SkM, Mb, and Mt (Figure 5a and Figure S8; Tables S4a and S5a). The only studied non-SkM tissues that have considerable expression of this gene are the thyroid gland (TPM, 10) and anterior cingulate cortex of the brain (TPM, 2.3) but SkM expression is much higher (TPM, 42; Table S4a). SkM, Mb, and Mt displayed a large region of strong promoter chromatin as well as intragenic enhancer chromatin. The SkM/Mb/Mt enhancer chromatin immediately downstream of the 3′ end of the gene overlapped a SkM-hypomethylated DMR and mouse cell-inferred MyoD binding sites in Mb and Mt [8,36]. Although heart and lung displayed negligible-to-low levels of *TMEM233* RNA (TPM, 0.3 and 1.2, respectively), they both exhibited some active promoter chromatin. However, they also had repressed chromatin as determined from their chromatin state based upon H3K27me3 enrichment. The high SkM and Mb specificity of this gene suggests a differentiation-associated function that needs to be maintained at many stages in this lineage.

Unlike *TMEM233*, *TMEM52* (Figure 5c) was highly expressed in several non-muscle tissues, namely, pancreas, adrenal gland, and testis, with slightly higher expression in pancreas than in SkM (TPM, 90 and 82, respectively). A SkM-associated calmodulin-like gene, *CALML6*, is located only 0.3 kb downstream of *TMEM52*. It too, is poorly characterized as to its function except for its BLASTP-deduced similarity to an EF-hand type of Ca$^{+2}$-binding protein [44] and upregulation in muscle from patients with myotonic dystrophy type 2 [45]. The gene bodies of *TMEM52* and *CALML6* cover most of a 6 kb super-enhancer in SkM and the pancreas. This super-enhancer is probably upregulating *TMEM52* much more than *CALML6* (e.g., SkM TPM for *TMEM52*, 82 versus for *CALM6*, 7). This is probably related to the finding that *TMEM52*, but not *CALM6*, exhibited strong promoter chromatin at the region around its TSS (Figure 5c; Table S4).

Although *TMEM38A* is expressed at moderately low levels in most examined tissues and cell cultures, its expression was highest in SkM (TPM, 249) and with the next-highest levels in most parts of the brain (TPM, 17–45) and heart (TPM, 17). These are the excitable tissues where TMEM38A, a trimeric intracellular cation channel-associated protein, has been proposed to be especially important [41]. SkM was the only tissue with a super-enhancer throughout most of its long intron 1. In addition, there was SkM-associated enhancer chromatin in the gene body of its nearest upstream neighbor, *SMIM7*, a gene encoding a membrane protein of uncertain function, which is expressed broadly and at much lower levels than *TMEM38A* in SkM (TPM, 7 versus 249), and at lower levels in SkM than in most other tissues (Table S4). This suggests that *TMEM38A* may have its expression in SkM enhanced not only by SkM-associated intragenic enhancer chromatin but also by *SMIM7* enhancer chromatin in SkM.

Both *TMEM38A*, highly specific for SkM, and the related *TMEM38B*, preferentially expressed in SkM but less specific for this tissue, have five exons, but with large differences in intron patterning that are reflected in different distributions of SkM enhancer chromatin at these genes (Figure 5b, d). The upstream neighbor of *TMEM38B* is the 0.7 kb intron-less *TAL2* gene, which encodes an oncogenic TF essential for normal brain development in mice and implicated in osteoclastogenesis [46,47]. Although *TAL2* has not been described as being related to SkM, we found that the highest (albeit low) steady-state levels of its RNA among the 53 studied tissues were in SkM (TPM, 0.7) and testis (TPM, 0.9; Table S4b and Figure S9). This tissue specificity is similar to that of *TMEM38B*. Mt also had a higher level of expression of *TAL2* (and *TMEM38A* and *TMEM38B*) than did non-myogenic cell cultures (Table S5b). This finding is consistent with a SkM function, not previously reported, for the TAL2 TF.

*2.6. Related Sequences of the Proteins Encoded by Some of the Studied SkM or SkM/Heart Genes Suggest Functional Similarities*

Most of the paralogs of the studied genes were in their respective families [34], as expected (Table S6). Paralogs to genes outside their families were particularly prominent for *LRRC14B*, a SkM- and heart-specific gene of poorly understood function, which is paralogous to many *PRAME*-family genes as well as to the broadly expressed *LRRC14*, a negative regulator of NFκB signaling [6,34]. PRAME is a testis- and melanoma-associated transcriptional repressor that inhibits retinoic acid signaling [48]. Searches for protein sequence similarity using the Basic Local Alignment Search Tool for proteins (BLASTP) revealed significant protein similarity between LRRC14B and both LRRC14 and PRAME. LRRC30 has significant aa sequence similarity to LRRC10, which, like LRRC30, is encoded by a very small gene with a single exon and expressed preferentially in a single type of tissue (heart). LRRC10 is a component of a cardiac voltage-gated L-type $Ca^{2+}$ channel whose mutation can cause dilated cardiomyopathy [49].

OSBPL6 and OSBPL11 proteins have been studied in non-muscle tissues where they are associated with cholesterol transport (Table S6). There is very high protein similarity between OSBPL11 and OSBPL10 (BLASTP E-value 0, and Query coverage, 90%), both of which are localized, in part, to the Golgi apparatus, but OSBPL11, and not OSBP10, is also found in the nucleus [50,51]. In addition, OSBPL11 has high similarity (BLASTP E-value, 1E-147, and Query coverage, 50%) to OSBPL9, with which it heterodimerizes. OSBPL6's protein structure very closely aligns with that of OSBPL3 (BLASTP E-value, 2E-127, and Query coverage, 80%), a protein that may regulate ER-morphology, the actin cytoskeleton, and aspects of cell adhesion, in addition to binding cholesterol [50,51].

The studied ASB proteins, ASB5 and ASB11, display very strong evolutionary conservation of their primary structure with each other and with ASB9 (Tables S6 and S7; [38]). These proteins have been implicated in regulating the size of the SkM and brain component (ASB5), gastrointestinal expansion and regenerative and embryonic myogenesis (zebrafish asb9/11), and spermatogenesis and ovarian granulosa cell maturation (ASB9) [38,52]. Unlike the evidence for the role of asb9/asb11 in myogenesis in zebrafish, *ASB9* and *ASB11* genes exhibited negligible expression in human Mb and Mt (Table S5c). ASB2 and ASB15, which show very high aa sequence similarity to each other, have been reported to be involved in myogenesis [53]. The aa sequence similarity of ASB4, ASB10, and ASB16, and that of ASB12 and ASB1 or ASB8 and ANKRD44, do not elucidate SkM functionality because of the lack of literature about the function of any of these genes in SkM other than the general associations of ASB proteins with protein ubiquitination, Notch signaling, suppression of cytokine signaling, and mitochondrial function [53].

Among the four examined *TMEM* genes, their encoded protein sequence comparisons were most informative for TMEM38A and TMEM38B. These two proteins are significantly similar to each other in aa sequences but not to other proteins (Tables S6 and S7). Both have been shown to be involved in SkM function or differentiation [42,54,55].

## 3. Discussion

SkM-upregulated genes in the *ASB*, *LRRC*, *OSBPL*, and *TMEM* gene families displayed SkM-associated enhancer chromatin that was usually intragenic and contained subregions exhibiting SkM DNA hypomethylation (Table 1). Our epigenetic findings for these genes are consistent with other studies indicating a central role for enhancers in determining or maintaining tissue-specific expression [16,23,56]. These 21 genes had been subject to little or no epigenetic analysis in SkM despite their likely or known roles in SkM development or maintenance and in diseases involving SkM or other tissues (e.g., Emery–Dreifuss muscular dystrophy, myotonic dystrophy type 1, osteogenesis imperfecta, and cancer; Table S6). Eight of these genes displayed SkM- and transcription-associated hypomethylated DMRs immediately upstream or downstream of the TSS (Table 1). Such promoter region DNA hypomethylation linked to tissue-specific gene expression was especially prevalent among *ASB* genes. Eight other genes had low methylation in the promoter region in all tissues but with broadening of the LMR in SkM. These findings are consistent with the importance of TSS-downstream DNA hypomethylation to transcription regulation [56].

The genes with SkM-associated promoter DNA hypomethylation showed very high specificity for SkM expression. For example, *LRRC30*, an unusually small (0.9 kb) gene that is transcribed essentially only in SkM, displayed a large SkM-specific region of promoter chromatin that covered the whole gene and was adjacent to hypomethylated enhancer chromatin upstream and downstream (Figure 2a). The function of the LRRC30 protein is uncertain but it has significant sequence similarity to LRRC10, which is encoded by another small (2.6 kb) intron-less gene that is a paralog of *LRRC30* (Tables S6 and S7). *LRRC10* is expressed predominantly in the heart and has a *LRRC30*-type pattern of promoter chromatin over the whole gene bordered by hypomethylated enhancer chromatin. However, for *LRRC10*, these epigenetic marks were seen in the heart (data not shown) rather than in SkM, as for *LRRC30*. This finding suggests that LRRC30 might have a similar function to that of the actin- and $\alpha$-actinin-binding of LRRC10 [34], except in SkM rather than in the heart.

The relative steady-state levels of RNA for the examined genes among different tissues were usually reflected in their total amount and distribution of intra- or inter-genic enhancer chromatin. Therefore, major contributions from tissue-specific post-transcriptional processing did not confound most of the relationships between tissue-specific transcription and tissue-specific RNA levels for the 21 studied genes. One of the best correlations of expression levels in SkM with epigenetics was with the presence of super-enhancers, consistent with their strong upregulation of expression in a tissue-specific or development-specific manner [27]. There may be contributions of tissue-specific epigenetics to isoform usage, possibly, for *OSBPL6* (Figure 3), whose encoded oxysterol binding-like protein might have a special function in SkM because of the importance of membrane cholesterol to excitation–contraction coupling and glucose transport [57].

*LRRC39*, a member of the *LRRC* super-family that encodes diverse proteins with leucine-rich repeat motifs [58], was unusual in its high expression specifically in SkM but only modest amounts of intragenic or promoter-adjacent enhancer chromatin in this tissue. This gene, which codes for a sarcomeric M-band protein, had only one 1.4 kb intragenic enhancer chromatin region (Figure 2d). Remarkably, in the 3′-UTR of the adjacent *DBT* gene, which is broadly but moderately expressed, there was enhancer chromatin specific to SkM, Mb, Mt, and the heart, all of which preferentially express *LRRC39*. We propose that *LRRC39* is using *DBT*-intragenic space to store an enhancer that contacts the *LRRC39* promoter, but not the *DBT* promoter, in the SkM lineage and in the heart. Surprisingly, Mt, which selectively express *LRRC39* at particularly high levels and for which we have CTCF binding profiles, did not show evidence of a potential CTCF insulator site between this *DBT*-resident enhancer and the *DBT* promoter (data not shown [35,59]). Such a CTCF site could have been responsible for preventing the *DBT* enhancer from upregulating *DBT* in SkM. In contrast, the SkM-associated super-enhancer that overlaps both *LRRC14B* and the 3′ end of the adjacent *PLEKHG4B* (Figure 1b) is probably prevented from interacting with the *PLEKHG4B* promoter by a strong constitutive CTCF binding site near the center of the *PLEKHG4B* 3′ UTR [35,59]. The interaction with the *LRRC14B* super-enhancer with the *LRRC14B* promoter rather than the *PLEKHG4B* promoter in SkM and the heart might also be aided by the unusually strong SkM/heart-specific expression of a 0.6

kb lncRNA gene 4 kb downstream of this promoter (Figure 2b). Like the *LRRC39-DBT* and *LRRC14B-PLEKHG4B* gene-pairs, *ASB10-IQCA1L* was a gene-pair where the enhancer in the 3′ end of one gene is likely upregulating the 5′ juxtaposed promoter of the other gene (Figure S7).

In a different epigenetic strategy involving neighboring genes, *TMEM52* and *ASB8* were juxtaposed tail-to-tail with *CALM6* and *PFKM* respectively, with which they shared a super-enhancer that appears to upregulate one of the genes in the gene-pair more than the other (Figure 5c and Figure S2). *CALML6* is a calmodulin-like gene [44] with no descriptive gene summary in Entrez Gene [34]. It is expressed preferentially in SkM, like its *TMEM52* neighbor. *PFKM* encodes a well-characterized SkM-associated phosphofructokinase that is expressed preferentially and at extremely high levels in SkM. Like *PFKM*, *ASB8* is transcribed at considerable levels in diverse non-muscle tissues, although it is expressed at the highest levels in SkM and is broadly expressed in cell cultures (Tables S4 and S5). Therefore, similar to *PFKM*, *ASB8* probably has a constitutive function as well as being needed at much higher levels in SkM than in other tissues.

Most of the 18 members of the chordate-specific *ASB* (Ankyrin Repeat and SOCS Box) gene family encode a specificity subunit of an E3 ubiquitin ligase complex [39,60], and half of these genes are preferentially expressed in SkM or, less frequently, in SkM and the heart (Table S4). The ankyrin repeats of this subunit determine specific protein–protein interactions that direct ubiquitination with subsequent proteosomal degradation of the target. *ASB2* is the most well studied of the nine SkM- or SkM/heart-associated *ASB* genes. Its predominant isoform in SkM, *ASB2β* (Figure 4d), encodes a protein implicated in regulating myogenesis, decreasing SkM mass, and possibly aiding recovery of SkM after exercise and contributing to the symptoms of myotonic dystrophy type 1 [53,60–63]. Despite the importance of the β or α isoforms of this gene to myogenesis, hematopoiesis, cardiomyocyte maturation, cardiac homeostasis, certain types of leukemia, and Notch, TGFβ, and retinoic acid signaling pathways (Table S6), there has been only one study published about epigenetic regulation of *ASB2*. That report described the *ASB2α* hematopoiesis-associated proximal promoter as upregulated by retinoid receptors [64]. Our analysis indicates that the very strong expression of *ASB2* specifically in SkM involves the hypomethylated DMR-containing super-enhancer spanning the gene and its strong promoter.

Unlike the high expression of *ASB2* in SkM, there was negligible *ASB2* RNA in Mb and only low levels in Mt (Table S5). Similarly, there was more enhancer chromatin in Mt than Mb, although even Mb and tissues with negligible expression of *ASB2* (e.g., liver) had some intragenic enhancer chromatin. Probably, the repressed (H3K27me3-enriched) chromatin at the 5′ end of *ASB2* (Figure 4d) negated the liver enhancer chromatin. The upregulation of *Asb2* (β isoform) upon differentiation of C2C12 Mb to Mt may facilitate this differentiation by controlled proteolysis of filamin B, which can promote cytoskeletal rearrangements [65]. *ASB4* and *LRRC14B* also display modest expression in Mt and negligible amounts of their RNA in Mb, which suggests roles for their encoded proteins in myogenesis. ASB4 is involved transiently in the formation of the embryonic vasculature [66] and in vascular differentiation in placenta [60]. It has been proposed to play a role in human preeclampsia during pregnancy. The temporally restricted role of ASB4 in promoting the development of blood vessels in response to oxygen tension occurs partly through targeted degradation of the protein ID2 (Inhibitor of DNA-binding 2) [67] and supports the postulated role for ASB4 in myogenesis because ID2 negatively controls myogenesis as well as vascular differentiation [34].

The genes encoding sarcomere-associated LRRC39 and sarcoplasmic reticulum- and ion-channel-associated TMEM38B showed very large increases in expression in Mt versus Mb (Tables S5 and S6), suggesting the importance of these structural proteins even at the Mt stage. In contrast, *ASB10*, *ASB11*, *ASB12*, *ASB15*, *LRRC30*, and *TMEM52* displayed negligible expression in both Mb and Mt as well as repressed chromatin state profiles, indicating that they are unlikely to play a role in human myogenesis, although zebrafish and mouse homologues of *ASB11* (*d-asb/9/11*) and *ASB15* were reported to be expressed in myogenic progenitor cells [38,68]. We found that *ASB5* had very high expression in Mb and Mt (Table S5) and very much enhancer chromatin in these cells (Figures 4c). These results are consistent with a previously postulated role for murine Asb5 in postnatal regenerative myogenesis, including at the muscle satellite cell stage [61] (Table S6). Similarly, Asb5

has also been implicated in the initiation of artery formation [69], and its gene is moderately expressed in a few smooth muscle-rich organs with concomitant transcription-favoring epigenetic marks (Figure 4c).

Although all the examined genes in the *ASB*, *LRRC*, *OSBPL*, and *TMEM* families had at least five-fold higher levels of RNA in SkM than in the median of the other examined 52 postnatal tissues, most were not exclusively expressed in the SkM lineage. *LRRC2* is preferentially expressed in ESC as well as in Mb and Mt and its overlapping *LRRC2-AS1* gene, only in ESC (Figure 2b; Table S5). These findings suggest that *LRRC2*, a gene already implicated in SkM and cardiac function, has a previously unreported role to play in ESC, in which its upregulated expression is similar to that of its inton-1 ncRNA gene, *LRRC2-AS1*. *LRRC20*, a little-studied gene, is transcribed at considerable levels in almost all studied tissues (Table S4) and, therefore, is likely to perform a constitutive function, although the much higher steady-state levels of *LRRC20* RNA in SkM suggest extra requirements for LRRC20 protein in this tissue. Genes of uncertain functionality in SkM, like *ASB10*, *ASB11*, *ASB15*, *LRRC2*, *LRRC14B*, and *TMEM38B*, have a strong specificity for expression in both SkM and the heart, although expression in SkM was higher. This expression pattern suggests that the encoded proteins have roles to play specifically in both skeletal and cardiac striated muscle, as for the above-mentioned SkM- and heart-associated sarcomeric protein LRRC39 [29]. In addition, *ASB2*, has considerable expression in heart although it meets our criteria for a SkM gene. *ASB2* is abnormally downregulated in a mouse model for hypertrophic cardiomyopathy with an associated buildup of desmin levels due to loss of ASB2-directed proteosomal-mediated degradation in cardiomyocytes [70]. Similarly, ASB2 can be a negative regulator of SkM mass [71].

Two paralogs, *TMEM38A* and *TMEM38B*, which encode intracellular multi-pass membrane proteins with high aa similarity, differ in the degree of their specificity for expression in SkM (Figure 5b, d; Table S4a), and this is related to their functionality. *TMEM38A* is not only more specific for SkM than is *TMEM38B*, but also is expressed at almost 10-fold higher levels in that tissue than is *TMEM38B*. TMEM38A (TRIC-A) forms sarcoplasmic reticulum channels in SkM distinct from those of TMEM38B (TRIC-B) and has $K^+$ gating properties more suitable for balancing $Ca^{+2}$ movements in SkM, a tissue with high demands for excitation–contraction coupling [41]. Explaining the specificity and extent of SkM expression of *TMEM38A*, the 5′ end of this gene overlaps a SkM-specific 20 kb super-enhancer and has a region of extended low DNA methylation, specifically in SkM. The lower SkM tissue-specificity of *TMEM38B* expression and the apparently more important roles of TMEM38B in non-muscle tissues are reflected in smaller regions of SkM enhancer chromatin and their location further from the TSS. Moreover, *Tmem38b* knockout mice have lethal respiratory failure while *Tmem38a* knock-out mice have a subtle phenotype involving SkM [72]. Also consistent with this difference in tissue-specificity of these two genes is the finding that a mutation in *TMEM38B*, but not *TMEM38A*, is associated with the bone-fragility disease, osteogenesis imperfecta [34].

The strong correlations of enhancer chromatin and DNA hypomethylation with tissue-specific expression patterns were also seen in non-muscle tissues for some of these SkM-associated genes. For example, *ASB4*, *LRRC38*, and *TMEM52* are preferentially expressed in the adrenal gland, in which they might function in development or homeostasis [73,74]. *ASB4* was also upregulated in the pituitary gland. The preferential expression of *TMEM52* in the pancreas as well as the adrenal gland was accompanied by enhancer/promoter or DNA hypomethylation patterns that are expected to favor expression in those secretory organs (Figures 1a,4b and 5c). The preferential expression of *ASB4* and *LRRC38* in SkM as well as in secretary organs may be related to the corresponding proteins' possible roles in secretion of glucocorticoids, energy homeostasis, and SkM catabolism under normal or disease conditions [75,76]. It might also be the result of common structures or functions in different tissues, such as the association of LRRC38-containing BK channels with excitable tissues as well as secretory organs [30]. The variety in the tissue and temporal developmental specificity in expression of the SkM-associated *ASB*, *TMEM*, *LRRC*, and *OSBPL* family members can explain the need for the precise tissue-specific targeting of epigenetic marks that we observed.

## 4. Materials and Methods

Quantification of RNA levels for tissues used the GTEx RNA-seq database [6]. Median TPM are given from the analysis of hundreds of samples for each of the 53 tissue types. Selection of genes specifically expressed in SkM from these GTEx data used the scheme outlined in Table S1. RNA-seq data for comparisons of GM12878 (a lymphoblastoid cell line) and the following cell strains, Mb, ESC, human umbilical vein endothelial cells (HUVEC), NHEK (normal human epidermal keratinocytes, and NHLF (normal human lung fibroblasts) were from transcription levels assayed by RNA-seq from ENCODE (Wold Lab) [35]. In the RNase-seq tracks for cell cultures, vertical viewing ranges were 0–4 for Figures 1–3, Figures S5 and S7, and 0–8 for Figures 4, 5 and Figure S6. For quantitation of cell culture RNA-seq data in Table S5, we determined the signal for each gene with Cufflinks [77]. For comparisons of RNA levels in Mb and Mt, we used our previously generated data in collaboration with Greg Crawford on well-characterized Mb and Mt [10,11]. All genome data coordinates refer to hg19/GRCh37.

The Roadmap Epigenomics chromatin state segmentation analysis (chromHMM, AuxilliaryHMM; 18 states; genome.ucsc.edu/cgi-bin/hgTrackUi?hgsid=791414343_JYuPYCGT8qAnmJs3tXDG1G57yFBH&c=chr6&g=hub_24125_RoadmapConsolidatedAssaya27004) [23,35] was used for determination of chromatin states (promoter, enhancer, repressed, etc.). The color code for chromatin state segmentation in the figures was slightly simplified from the original, as shown in the color keys in the figures. Chromatin state segmentation data are not available for postnatal adrenal gland or skin for comparison to the bisulfite-seq data. Bisulfite-seq profiles of genome-wide DNA methylation and DNaseI-hypersensitivity profiling (Figures S5–S7) were from RoadMap data [23]. SkM and heart refer to left ventricle and psoas muscle. Further descriptions of muscle samples used for these epigenetic analyses were given previously [11,23]. Super-enhancers were assessed by dbSUPER [26] and by inspection of chromatin segmentation state tracks as well as H3K27ac tracks. For identification of potential MYOD binding sites, orthologous sequences to murine C2C12 Mb and Mt binding sites from MyoD ChIP-seq [36] were mapped to the human genome (hg19). Bisulfite-seq data [23,35] for psoas (SkM), left ventricle, aorta, C14+ monocytes (mislabeled "macrophages" in the UCSC Genome Browser), lung, and adipose (subcutaneous) were used to determine significant SkM DMRs and heart DMRs by our previously described stringent method [78,79]. In brief, this DMR determination involved applying Fisher's exact test to the counts of methylated and unmethylated reads in each sample to produce site-specific *p*-values and then determining the joint probability of a sequence of five or more consecutive *p*-values to identify statistically significant regions at the 0.05 level with subsequent filtration to include only those regions with an average percent methylation difference of at least ±20%, length greater than 250 bp, and containing no gaps between consecutive sites greater than 200 bp. LMRs, which are also shown in the figures, refer to regions with significantly lower DNA methylation in a given region than in the rest of the same genome, as determined by Song et al. [28].

Paralogs were those listed in Genecards (https://www.genecards.org/) from the Ensembl database (https://www.ensembl.org). Protein BLAST analyses were done using the full-length sequences for the proteins listed in Table S6 as query sequences using the National Center for Biological Information website: https://blast.ncbi.nlm.nih.gov/Blast.cgi?PROGRAM=blastp&PAGE_TYPE=BlastSearch&LINK_LOC=blasthome. Only the highest scoring BLASTP results based on E-values are listed in Table S7.

## 5. Conclusions

Our analysis of the transcriptomics and epigenetics of a small number of genes all selectively expressed in SkM illustrates the detailed insights into cis-acting transcription regulatory regions that can be obtained with the RoadMap [23] and GTEx [6] databases. The genome-wide, single-base resolution chromatin state maps and methylome data allowed us to identify tissue-specific differences in the location of enhancers and DMRs within a given gene, in nearby intergenic regions, and even in neighboring genes that can help explain the tissue-specificity of expression of a given

SkM-associated gene. These results can inform studies of altered expression with exercise**,** aging**,** or disease by broadening our perspective of the regulatory regions studied from the frequently addressed core promoter and immediate-upstream enhancers. In addition**,** we were able to gain insights into different epigenetic strategies used by genes in a given family to effect precise patterns of tissue and cell-type specificity and to begin to understand the functioning of some little-studied SkM-associated genes in these families.

**Supplementary Materials:** The following are available online at www.mdpi.com/xxx/s1, Figure S1: GTEx violin plots for *LRRC38* and *LRRC14B* expression for each tissue type shown in Figure 1 in the main text. Figure S2. GTEx violin plots for *LRRC30*, *LRRC2*, *LRRC29* and *LRRC39* expression for each tissue type shown in Figure 2 in the main text. Figure S3. GTEx violin plots for *OSBPL6* and *OSBPL11* expression for each tissue type shown in Figure 3 in the main text. Figure S4. GTEx violin plots for *ASB16*, *ASB4*, *ASB5* and *ASB2* expression for each tissue type shown in Figure 4 in the main text. Figure S5: *ASB12* (ankyrin repeat and SOCS box containing 12), a little-studied SkM-specific gene, displays SkM-specific promoter-region hypomethylation like *ASB16*, *ASB4*, *ASB5*, and *ASB2*. Figure S6: *ASB8* (Ankyrin repeat and SOCS Box Containing 8), and its adjacent gene *PFKM* (Phosphofructokinase, Muscle) are expressed at the highest levels in SkM and share a super-enhancer. Figure S7: *ASB10* (Ankyrin repeat and SOCS Box Containing 10), which is preferentially expressed in SkM and heart, overlaps SkM/heart-specific enhancer-and-promoter chromatin containing SkM and heart hypomethylated DMRs. Figure S8: GTEx violin plots for *TMEM233*, *TMEM38A*, *TMEM52* and *TMEM38B* expression for each tissue type shown in Figure 5. Figure S9: GTEx data for *TAL2* expression levels. Table S1: Selection of SkM preferentially expressed genes and gene families for epigenetic analysis. Table S2 Skeletal muscle-preferentially expressed genes. Table S3: SkM-preferentially expressed genes that had little or no description of their function in SkM based on PubMed citations. Table S4: Tissue expression levels of 21 SkM or SkM/heart *ASB*, *LRRC*, *OSBPL*, and *TMEM* genes and some of their neighboring genes as well as known disease relationships for these studied genes. Table S5: Expression of SkM or SkM/heart genes listed in Table S4 in myoblasts (Mb) and myotubes (Mt) and other cell cultures. Table S6: Paralogs, Protein BLAST comparisons, and gene function summaries for SkM and SkM/heart *ASB*, *LRRC*, *OSBPL*, and *TMEM* family genes. Table S7: Protein BLASTP searches for examined genes that are preferentially expressed in SkM or SkM and heart (summarized in Table S6).

**Author Contributions:** K.C.E. conceived the study. K.C.E. and M.E. did the bioinformatics research for the study and wrote the manuscript. M.L. did the determination of DMRs for the study.

**Funding:** This research was supported in part by grants to ME from the National Institutes of Health (NS04885), the National Institutes of Health (National Center for Advancing Translational Sciences, award number UL1TR001417), and the Louisiana Cancer Center.

**Acknowledgments:** We thank Melody Badoo for help with Cufflinks under the auspices of COBRE grant NIGMS P20GM103518.

**Conflicts of Interest:** The authors declare no conflicts of interest.

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
