# Peer review of "Epigenetics of Skeletal Muscle-Associated Genes in the ASB, LRRC, TMEM, and OSBPL Gene Families"

_2075-4655_

Round 1
Reviewer 1 Report
This manuscript aims to identify genes specifically expressed in skeletal muscle (SkM). The authors successfully leveraged existing data from Epigenome roadmap and GTex and found genes in four family members with little or unknown role in SkM function. Possible regulation of gene expression by DNA methylation and chromatin state have been extensively discussed. Overall it is an interesting paper that is clearly written.
Major points:
The authors used chromatin state algorithms determined by roadmap consortium and referred to them as “characteristic histone modifications”. Having these characteristic marks laid out somewhere in the paper would help the readers to understand better how the promoters/enhancer/etc were determined by the authors. This also applies to the “intragenic enhancer chromatin”, “intergenic proximal enhancer”, and “intergenic distal enhancer chromatin” in table 1. Gene expression levels are determined using GTex data. However, how significant the differences are between TPM values from difference tissues are not clearly defined. In other words, is TPM 238 really different from TPM of 302? 0.7 is different from 0.9?
Minor points:
In all figures, please change the color of repressed chromatin from brown-red to something else. It is similar to red, which marks promoter regions. Table 4a is missing.
Author Response
Reviewer 1
This manuscript aims to identify genes specifically expressed in skeletal muscle (SkM).
The authors successfully leveraged existing data from Epigenome roadmap and GTex and found genes in four family members with little or unknown role in SkM function. Possible regulation of gene expression by DNA methylation and chromatin state have been extensively discussed.
Overall it is an interesting paper that is clearly written.
Major points:
The authors used chromatin state algorithms determined by roadmap consortium and referred to them as “characteristic histone modifications”. Having these characteristic marks laid out somewhere in the paper would help the readers to understand better how the promoters/enhancer/etc were determined by the authors. This also applies to the “intragenic enhancer chromatin”, “intergenic proximal enhancer”, and “intergenic distal enhancer chromatin” in table 1.
Response to Reviewer 1
1a. We have clarified the nature of the histone modifications that are used in the algorithms for Roadmap chromatin state segmentation by adding the underlined words below to Results.
L. 137 -140. Chromatin states were determined by the Roadmap Epigenomics Consortium [23] from genome-wide profiles of histone H3 lysine-4 trimethylation (H3K4me3) and H3 K27 acetylation (H3K27ac) for promoter chromatin and H3K4me1 and H3K27ac for enhancer chromatin (18-State/Auxiliary Hidden Markov Model).
L. 361. However, they also had repressed chromatin, as determined from the chromatin being enriched in H3K27me3.
In the original manuscript, we already had the following statement.
Probably the repressed (H3K27me3-enriched) chromatin at the 5’ end of ASB2 (Figure 4d) negated the liver enhancer chromatin.
In the footnote to Table 1, we added the parenthetical explanations given below.
aThe epigenetic features for these 21 genes are shown in Figures 1-5 and S5-S7, except for ASB11 and ASB15; chrom, chromatin; assoc, associated; prom, promoter (H3K4me3/H3K27ac-enriched); enh, enhancer (H3K4me1/H3K27ac- enriched), constit, constitutive (present in all or almost all tissues); LMR, low methylated region (determined by the local bisulfite-seq profile relative to the whole genomic profile of the given tissue [28])
1b. In the first sentence of the second paragraph of methods, we added the URL shown below for further clarification (L 625).
The Roadmap Epigenomics chromatin state segmentation analysis (chromHMM, AuxilliaryHMM; 18 states (https;\\genome.ucsc.edu/cgi- bin/hgTrackUi?hgsid=791414343_JYuPYCGT8qAnmJs3tXDG1G57yFBH&c=chr6&g=hub_24125_RoadmapC onsolidatedAssaya27004 [23, 35] ) was used for determination of chromatin states (promoter, enhancer, repressed, etc.).
1c. We have added the following to the first footnote to Table 1.
intragenic enh chromatin, enhancer chromatin overlapping the gene body; intergenic proximal enh chrom, enhancer chromatin upstream of and adjacent to the promoter region; intergenic distal enh chromatin, enhancer chromatin far upstream of the promoter region
Reviewer 1. Gene expression levels are determined using GTex data. However, how significant the differences are between TPM values from difference tissues are not clearly defined. In other words, is TPM 238 really different from TPM of 302? 0.7 is different from 0.9?
Response to Reviewer 1
- Because the data for almost all of the GTEx tissue types and for all of the ones used in Figures in this manuscript come from 100s of individual samples, large differences in TPM are convincing.
We did not try to explain any small differences in gene expression such as Reviewer 1 posits.
- We chose genes with SkM TPMs that have 5-fold greater TPM in skeletal muscle relative to the median TPM of the set of 53 non-SkM samples (GTEx, RNA-seq database), as stated in Results. With these stringent criteria we have found some genes in important gene families that have been largely overlooked by previous authors who have sampled the databases for genes important for skeletal muscle function. We have added to Table S4, the GTEx sample number for each of the tissue types, which happens to be highest for SkM (803 biological replicates).
- Importantly, for expression/epigenetic correlations, we have considered the tissue-specific epigenetics only of tissues that had large differences in expression of the given gene, as assessed by median TPM values for each sample in GTEx. To emphasize this point, we have added five supplementary figures (Figures S1 - S4 and S8) that show violin plots giving the interquartile range of TPM values as well as the median TPM and outliers for all the tissues and corresponding genes shown in Figures 1 - 5. The significance of these tissue-specific differences in TPM is demonstrated by the lack of overlap of the interquartile range of TPM from hundreds of samples per tissue. Where there is overlap for the interquartile range of TPM of a given gene between two tissues, we refer to these genes as having similar expression levels in both tissues, e.g., for LRRC14B in SkM and heart, which is designated a SkM/heart gene rather than a SkM gene. We added the following underlined items to Results to clarify this issue of how large the tissue-specific differences in expression were that we are considering in correlations with tissue-specific epigenetics.
L. 143-145. With the exception of heart and adrenal gland, as described below, the non-SkM tissues examined for correlating tissue-specific epigenetics with transcription in Figures 1 and 2 had no overlap of their interquartile range of TPM values with that of SkM (Figures S1 and S2; Table S4a).
L. 264. The greater number or extent of enhancer chromatin regions in SkM than in other tissues is in accord with their tissue-specific expression profiling (Figures 3 and S3).
L. 291. Similarly, for the tissue types shown in Figure 4, the interquartile range of TPM values for SkM was higher than those of the other examined tissue types with no overlaps except for ASB4 in SkM gland and pituitary (Figures 4b and S4).
L352. TMEM233 is highly and specifically expressed in SkM, Mb, and Mt (Figures 5a and S8; Tables S4a and S5a).
Reviewer 1. Minor points:
In all figures, please change the color of repressed chromatin from brown-red to something else. It is similar to red, which marks promoter regions. Table 4a is missing.
Response to Reviewer 1
3a. We have changed the color of repressed chromatin to brown from brown-red as requested. We have also added more description of the types of promoter and repressed chromatin to the legend for Figure 1.
Chromatin state segmentation denotes predicted promoter chromatin (prom; light or bright red as weak or strong promoter, respectively), strong enhancer (enh; orange), weak enhancer (wk enh; yellow), or repressed chromatin (rep: dark or light gray, enriched in H3K27me3; light blue or blue-green, enriched in H3K9me3; brown, bivalent promoter enriched in H3K27me3 and H3K4me3) or H3K36me3-enriched transcribed chromatin (txn chrom, green).
3b. Table 4a was not missing. Supplemental Table 4 was divided into two parts previously labeled as (a) and (b). We have changed the heading and subheadings to now read Table S4a and Table S4b.
Reviewer 2 Report
Epigenetics of skeletal muscle-associated genes in the ASB, LRRC, TMEM, and OSBPL gene families manuscript is well written and may be of interest for scientific community. Deeper understanding and extensive research is necessary but beyond scope of this publication.
Author Response
Response to Reviewer 2.
No changes were suggested except for the item “Introduction can be improved” being checked. As to the Introduction, we already have 79 references in this manuscript itself and many more in the Supplementary Tables. With respect to further research in the future, part of the rationale for looking at transcription and epigenetic correlations in depth using in vivo data from humans, as we did, is to lay an in vivo-derived groundwork for future research.
Reviewer 3 Report
In the manuscript entitled "Epigenetics of skeletal muscle-associated genes in the ASB, LRRC, TMEM, and OSBPL gene families" by Ehrlich et al., the authors describe results of transcriptomic and epigenomic data of 21 genes which are specifically expressed in SkM. Data from publically available databases were used for these analyses. Tissue specific differences in the location of enhancers and methylation profiles within a given gene, in nearby intergenic regions and in neighboring genes were identified. Overall, data in the main manuscript and in the supplementary files are well presented and described.
Comments:
1. Line 36: Ca+2 should be Ca2+
2. In legend to Figure 1 the authors mention "four cell cultures", however, in panel A only 3 cell lines are mentioned and in panel B the number of cell lines is unclear.
3. Methods: Description of bioinformatic data analyses is almost missing, for the reader it is unclear which type of data were obtained from the databases, if some data were raw data that needed to be aligned, which version of the genome was used, which cut-off for differential methylation were defined...
4. Are the selected genes disease-relevant?
Author Response
Reviewer 3
In the manuscript entitled "Epigenetics of skeletal muscle-associated genes in the ASB, LRRC, TMEM, and OSBPL gene families" by Ehrlich et al., the authors describe results of transcriptomic and epigenomic data of 21 genes which are specifically expressed in SkM. Data from publically available databases were used for these analyses. Tissue specific differences in the location of enhancers and methylation profiles within a given gene, in nearby intergenic regions and in neighboring genes were identified.
Overall, data in the main manuscript and in the supplementary files are well presented and described.
Comments:
Reviewer 3. Line 36: Ca+2 should be Ca2+
Response to Reviewer 3
This change has been made.
Reviewer 3. In legend to Figure 1 the authors mention "four cell cultures", however, in panel A only 3 cell lines are mentioned and in panel B the number of cell lines is unclear.
Response to Reviewer 3
Only three cell cultures were used for the overlaid signal in these tracks for both panels as we now state in the Figure 1 legend in response to this reviewer.
Reviewer 3. Methods: Description of bioinformatic data analyses is almost missing, for the reader it is unclear which type of data were obtained from the databases, if some data were raw data that needed to be aligned, which version of the genome was used, which cut-off for differential methylation were defined...
Response to Reviewer 3
3a. All data were obtained from publically available databases with the exception of skeletal muscle and heart DMRs and MyoD binding sites. Details about our use of the public databases were given in the first 20 lines of Materials and Methods in the original manuscript followed by more text about our own database tracks.
3b. No raw data needed to be aligned with the exception of MyoD binding as described in Response 3c, below. In the legend to Figure 1 in the original manuscript, we stated “All tracks are from hg19 in the UCSC Genome Browser and are aligned.” In the revised manuscript, we now reiterate the use of hg19 in Materials and Methods.
623. All genome data coordinates refer to hg19/GRCh37.
3c. The MyoD binding site data were previously described but we have now added “hg19” to the following sentence.
635. For identification of potential MYOD binding sites, orthologous sequences to murine C2C12 Mb and Mt binding sites from MyoD ChIP-seq [36] were mapped to the human genome (hg19).
3d. We did not use an arbitrary cut-off for differential DNA methylation but rather we based our DMRs on a very stringent methodology, which was devised by mathematician coauthor Michelle Lacey and which we reference in Materials and Methods. This method has been described in detail in our previous (and cited) references: [78,79] and was described as shown below in the original manuscript.
Bisulfite-seq data [23, 35] for psoas (SkM), left ventricle, aorta, C14+ monocytes (mislabeled “macrophages” in the UCSC Genome Browser), lung, and adipose (subcutaneous) were used to determine significant SkM DMRs and heart DMRs by our previously described stringent method [78, 79].
To give extra information about our previously published methodology, we have added the following sentences to Materials and Methods.
In brief, this DMR determination involved applying Fisher’s exact test to the counts of methylated and unmethylated reads in each sample to produce site-specific p-values and then determining the joint probability of a sequence of five or more consecutive p-values to identify statistically significant regions at the 0.05 level with subsequent filtration to include only those regions with an average percent methylation difference of at least ±20%, length greater than 250 bp, and containing no gaps between consecutive sites greater than 200 bp.
In Results, the first time DMRs are mentioned we stated the following in the original text, which we retain in the revised manuscript.
We determined significant SkM-specific differentially methylated regions (DMRs) from bisulfite- seq data [23] for SkM vs. for heart, aorta, lung, adipose tissue, and monocytes using stringent criteria (see Methods).
Reviewer 3. Are the selected genes disease-relevant?
Response to Reviewer 3
We had noted in the original Results text that TMEM38B is associated with osteogenesis imperfecta, which can include SkM and cardiac deficiencies and referred the reader to Table S6.
In the original manuscript, we already had extensive disease associations to the studied genes in a column in Table S6 with the heading “Known function of protein encoded by query gene (disease associations in boldface).” In addition, that table in the original manuscript gave references for the disease associations.
To emphasize these disease associations, we have added to the end of the first paragraph in Results the following statement.
Six of the studied genes have been associated with cancer, two with cardiac disease (LRRC2 and LRRC14B), one with Emery-Dreifuss muscular dystrophy (TMEM38A), one with myotonic dystrophy type 1 (ASB2), one with glaucoma (ASB10), one with Treacher Collins syndrome (a craniofacial disease; OSBPL11), and one with osteogenesis imperfecta (TMEM38B; Table S6).
We also inserted a short sentence into the Abstract to highlight the important point raised by this Reviewer.
Some of these genes have associations with SkM or heart disease, cancer, bone disease, or other diseases.
In addition, we added to the first footnote in Table S4 the following:
See Table S6 for brief descriptions of function and disease relationships of the genes in Part a of this table.
Round 2
Reviewer 1 Report
The authors adequately addressed my points.